# Mothers' sleep deficits and cognitive performance: Moderation by stress and age

**Kirby Deater-Deckard**[1], **Mamatha Chary**[1]*, **Maureen E. McQuillan**[2], **Angela D. Staples**[3], **John E. Bates**[4]

**1** University of Massachusetts at Amherst, Amherst, Massachusetts, United States of America, **2** Indiana University School of Medicine, Indianapolis, Indiana, United States of America, **3** Eastern Michigan University, Ypsilanti, Michigan, United States of America, **4** Indiana University, Bloomington, Indiana, United States of America

☯ These authors contributed equally to this work.
* mchary@umass.edu

**Data Availability Statement:** All relevant data are within the paper and its Supporting information files.

**Funding:** The study has been funded by Grants MH099437 from the National Institute of Mental

## Abstract

There are well-known associations between stress, poor sleep, and cognitive deficits, but little is known about their interactive effects, which the present study explored in a sample of mothers of toddlers. Since certain types of cognitive decline start during the 20s and continue into later ages, we also explored whether mothers' age interacted with stress and sleep in the prediction of cognitive functioning. We hypothesized that poorer sleep [measured using one week of 24-hour wrist actigraphy data] and having more chronic stressors [e.g., life events, household chaos, work/family role conflict] would be linked with poorer cognitive performance [both executive function and standardized cognitive ability tasks], and that the interactive combination of poorer sleep and more stressors would account for the effect. We also explored whether this process operated differently for younger versus older women. In a socioeconomically and geographically diverse community sample of 227 women with toddler-age children [age, $M = 32.73$ yrs, $SD = 5.15$ yrs], poorer cognitive performance was predicted by greater activity during the sleep period, shorter sleep duration, and lower night-to-night consistency in sleep; it was not associated with higher levels of stress. The interactive effects hypothesis was supported for sleep activity [fragmented sleep] and sleep timing [when mothers went to bed]. The combination of more exposure to stressors and frequent night waking was particularly deleterious for older women's performance. For younger women, going to bed late was associated with poorer performance if they were experiencing high levels of stress; for those experiencing low levels of stress, going to bed late was associated with better performance.

## Introduction

Cognitive performance is related to adaptive function in many spheres of life such as completion of goal-directed behavior, regulation in interpersonal relationships, and various aspects of behavioral self-control [1], so it is important to understand how differences in cognition arise. Deficits in cognitive functioning among healthy individuals are more prevalent among those who have poorer sleep and those who face more chronic stressors [2, 3]. Poor sleep degrades

Health and HD073202 from the Eunice Kennedy
Shriver National Institute of Child Health and
Human Development.

**Competing interests:** The authors have declared
that no competing interests exist.

cognitive processing speed, efficiency, and accuracy by reducing alertness and vigilance [4, 5].
Exposure to chronic stressors also negatively affects cognition by taking up cognitive resources
and presenting persistent and distracting concerns [5–7]. These findings raise the question of
whether the stress and sleep predictors independently or interactively predict variance in cog-
nitive functioning.

Another relevant question is whether effects of stress and sleep are further modulated by
age differences, which have been shown in some studies of cognitive functioning [8, 9]. There
is considerable heterogeneity in cognitive functioning across the lifespan, with certain types of
cognitive abilities beginning to decline in the 20s and some not beginning noticeable decline
until later ages [10]. In the current study, we addressed these questions by focusing on mothers
in their 20s through 40s- since for many women, these years are marked by the entry into par-
enthood [i.e., in the United States, nearly 90% have a child by the time they are 44; 11], which
results in both disrupted sleep and higher stress due in part to childrearing demands [12, 13].
While extensive literature has covered changes in mothers' postpartum sleep, fewer studies
have examined sleep in mothers of toddlers, after the postpartum period. Toddlerhood is a
period of time characterized by a number of normative changes in child sleep which could
influence mothers' sleep including decreases in daytime sleep [14], decreases in night awaken-
ings [15], and decreases in the total amount of sleep [16]. These years are also marked by high
levels of stress for mothers, due to factors such as having to create and maintain a work/home
balance, face increases in negativity/oppositionality in child behavior, and manage daily par-
enting "hassles" [17]. Mothers are an understudied population in the research on sleep and
stress effects on cognition, with even less research taking into account maternal age
differences.

## Stress, sleep and cognition

Exposure to chronic stressors leads to decrements in cognitive functioning and sleep [18–20]
via complex processes that disrupt circadian hormones that regulate the stress response and
sleep-wake cycle [21–23]. Chronic stress also increases allostatic load; the "wear and tear" on
the body's psychological and physiological systems involved in stress reactivity and regulation
[24]. Stress reallocates finite executive control resources to deal with the stressor at hand, thus
impairing top-down cognitive processes such as executive function [6].

Exposure to daily stress is also associated with high levels of day-to-day sleep variability in
terms of duration and timing [25]. For example, individuals who experience high levels of
daily stress may have short and fragmented sleep on one night, but due to sleep deprivation
and sleepiness, may sleep longer on the next night. Thus, stress is associated with variability
in sleep duration and nighttime wakings. In a similar vein, poor sleep marked by short dura-
tions and nighttime wakings is also implicated in cognitive performance differences, due to
increases in fatigue and degradation in alertness and attention [3, 26]. Studies show that even
one night of many nighttime wakings is associated with poorer performance on sustained
attention and working memory tasks the next day [27]. Later sleep timing [which is more com-
mon among those experiencing daily stress; 28] may have associations with poorer memory
performance, in part through covarying morphological changes in the hippocampus [29].
Thus, research suggests that multiple aspects of sleep [e.g., timing, duration, consistency, wak-
ings/activity] should be considered, when examining sleep problems as they relate to stress
and cognition.

Women with young children are one subset of the population that face the intersection of
both poorer sleep and higher levels of stress. Mothers tend to have poorer sleep than non-par-
enting women due in part, to their own children's sleep disruptions [12]. In terms of stress, in

the current study of women with young children, we focused on inclusion of specific stressors that have been shown to be quite prevalent and influential for this segment of the population and are known risk factors for decrements in health and cognitive function. These include living in a poorly regulated and noisy or "chaotic" home environment [30], caring for a child with disruptive behavior problems such as aggression or inattention symptoms [13], being a single parent [31], from a lower socioeconomic bracket [32], facing adverse life events such as negative job or relationship changes [33], facing chronic daily parenting hassles [34], lacking access to social support from other adults [35], and experiencing "role overload" at home and work [36].

Normative aging throughout the lifespan also plays a key role in explaining effects of stress and sleep on cognitive performance. For women and men alike, from the late 20s onward, cognitive decline occurs in processes involved in effortful memory storage, manipulation, and use of information [including executive function- EF, described below as part of the cognitive self-regulation system], along with more automatic factors involving processing speed, verbal ability, and non-executive memory [37, 38]. In addition, sleep difficulties increase for women and men alike, starting from the early 20s [39]. Sharp declines are seen in slow wave sleep, sleep duration and uninterrupted sleep, reflected in poorer quality sleep and insomnia [18, 40, 41].

At the same time, for some individuals, development from young adulthood into older age is accompanied by positive adaptive growth—in psychological and physical coping mechanisms, goals, healthy behaviors, and personal contentment [e.g., exercise, stress management, social engagement, challenging cognitive engagement] that can offset age-related declines [42–44]. For example, there is evidence that older mothers tend to have higher levels of education and greater financial security [45, 46]. There is also evidence that older mothers report lower rates of conflict with children and higher self-efficacy in parenting [47, 48]. For these reasons, another goal of the current study was to examine whether the hypothesized interaction between sleep difficulties and stressors would operate differently for younger versus older women, spanning the 20s, 30s, and 40s.

## Study aims and hypotheses

Among adult women, poorer sleep covaries with higher exposure to chronic stressors [e.g., less social support and the many stressors that are correlated with lower socioeconomic status; 49, 50]. However, it is not known whether and how greater exposure to chronic stressors and poorer sleep work together [i.e., independently, or interactively] to contribute to decrements in cognitive performance—and, whether this operates differently for younger versus older adults. It is important to examine potential non-independent interactive effects between potential predictors, because information about independent effects of those predictors is incomplete and misleading if those predictors' effects are actually conditioned on the level[s] of the other predictors [51]. Based on the literature, our first hypothesis of "independent effects" was that poorer cognitive performance would be found for women who had both poorer sleep and more stressors [prior to testing for an interaction between sleep and stress]. Second, we hypothesized an interactive effect, whereby poorer sleep would be predictive of poorer cognitive performance, but *more so* for mothers who reported a higher number of stressors. Our third and final aim was to investigate whether that interactive effect operated similarly or differently for younger versus older women [i.e., a three-way interaction effect with maternal age]. For this third aim, two competing hypotheses seemed plausible given the research literature. On the one hand, the anticipated stress/sleep interactive effect might be *attenuated for older women* because they may have acquired a broader set of effective stress management strategies from experience and may be of higher SES [i.e., older age as a resiliency

factor]. On the other hand, the stress/sleep interaction effect might be *strongest for older women*, because of normative declines in stress regulation and cognitive systems that accompany typical aging [i.e., older age as a risk factor].

## Method

### Participants

The Institutional Review Boards at Virginia Polytechnic Institute and State University (IRB 12–811) and Indiana University (IRB 0811000120) approved the studies. Written consent was obtained by participants. The study included two community samples of women with 2.5-year-old toddlers in Indiana and Virginia. Recruitment was primarily accomplished through a database using county birth records and community outreach efforts, such as through the local Head Start agency and the Housing Authority. Advertisements were also used, such as postcards and flyers throughout the community. Transportation was offered and compensation was provided to all participants.

The present sample included 241 women with actigraphy and cognitive performance data. Participants were 21 to 50 years old [mean [*M*] = 32.75, standard deviation [*SD*] = 5.12]; their toddler-aged children were 2.5 years old on average [age range: 30.12–33.24 months; 51% female]. This was the only child in 30% of the families. Just over half of the families [57%] had two or three children in the home, and the remaining 13% had four or more children. Using information gathered on parent education and occupational prestige, the sample was middle class [range = 15.5–66, *M* = 48.81, *SD* = 17.76; Hollingshead, 1975; 56]. Just over one-quarter [28%] of the mothers worked 40 or more hours per week outside the home, just under half of the sample [41%] worked outside the home fewer than 40 hours per week, and the remaining third [31%] did not work outside the home. This is representative of the US population generally, in which about 70% of households with young children have a part-time or full-time employed mother [52]. Ninety percent of mothers were White, 4% were Hispanic/Latinx, 3% were African American/Black, 1% were Asian American, and 2% identified as mixed race, American Indian, or other. Eighty-four percent of the mothers were married, 4% were divorced, separated, or remarried, and the remaining 12% were single mothers who had not married the target child's biological father.

### Procedure

Women completed a survey and participated in home and lab visits. They wore a wrist actigraph for one week and visited the lab to complete a battery of cognitive tasks and fill out questionnaires.

### Measures

**Stressors.** There are many ways to operationalize chronic stressors. In the current study, we used a cumulative or multiple stressor index, to represent individual differences in exposure to a group of covarying chronic stressors that are relatively common for adult women. In this method, it is assumed that the numbers and severity of stressors all work together in a cumulative fashion to explain associations with other constructs. Cumulative risk indexes [CRI's] have been shown to be more predictive of negative outcomes, relative to the study of a single risk factor [53]. We preferred this approach to approaches that focus on a small number of stressors and examine each stressor in isolation [54]. Although there is not consensus about which stress variables should be examined, previous research has shown that the precise combination of stressors may not be as important compared to the mere choice to consider risk in

aggregate [55]. Indicators of interest included low SES [based on Hollingshead, 1975; 56], being a single parent [coded 0 = two-parent family, 1 = one-parent family], having more stressful life events, low social support, high levels of child misbehavior, more daily parenting task demands, higher household chaos, and greater role overload. This cumulative risk index was also used in the McQuillan et al., 2019 paper on the same sample of mothers [57].

On the Changes and Adjustments Questionnaire [CAQ; 58], mothers reported their stressful life events over the past year [e.g., moving, renovations, death in family] from a list of seventeen events; a total score was used [$M$ = 2.34 events, $SD$ = 2.27 events, $\alpha$ = .93].

On the Social Support scale [59], mothers rated their informational and companionship supports or lack of supports using a four-point Likert scale, ranging from 0 = *never* to 4 = *often*. The eleven lack of support items assessed the frequency of unwanted advice or intrusion, the failure of others to provide help, others' unsympathetic or insensitive behavior, and experiences of social rejection or neglect. An average score across these 11 items was computed [$M$ = 1.00, $SD$ = 0.63, $\alpha$ = .89].

To assess child misbehavior, the externalizing behavior scale of the Child Behavior Checklist [CBCL 1 ½—5; 60], the intensity scale of Eyberg Child Behavior Inventory [ECBI; 61], and the child misbehavior scale [62] were used. The CBCL externalizing scale was comprised of the sum of aggression, attention problems, and rule breaking items rated as 0 = *not true*, 1 = *somewhat or sometimes true*, and 2 = *very true or often true* [$M$ = 11.86, $SD$ = 7.29, $\alpha$ = .91]. The ECBI intensity scale asks mothers to use a 7-point Likert scale [1 = *never* to 7 = *always*] to report frequency of child oppositional behaviors, such as "dawdles in getting dressed" and "argues with parents about rules,". A summed score was computed [$M$ = 103.93, $SD$ = 22.79, $\alpha$ = .89]. The Child Misbehavior Scale assesses the frequency over the past six months of 12 misbehaviors such as "tantrums" and "hits/bothers adults". Items were rated on a three-point Likert scale, 0 = *does not happen*, 1 = *happens sometimes*, or 2 = *happens a lot*. The sum of these items was computed [$M$ = 7.31, $SD$ = 3.01, $\alpha$ = .72].

To assess parenting daily hassles, women completed the Parenting Daily Events scale [63]; the parenting tasks subscale was used [eight items]. Mothers rated the frequency [1 = *never* to 4 = *constantly*; range 0–32] and intensity [1 = *no hassle* to 5 = *big hassle*; range: 0–40] of common parenting demands [e.g., cleaning messes, managing schedules]. The sum of the ratings of frequency and intensity for these eight items was computed for a final score [$M$ = 40.31, $SD$ = 8.39, $\alpha$ = .81].

The level of household chaos was reported using the Confusion, Hubbub, and Order Scale [CHAOS, 64]. Women responded to 12 binary [1 = *yes*, 0 = *no*] indicators [e.g., "You can't hear yourself think in our home", "It's a real zoo in our home"]. These were summed to form a chaos score [$M$ = 3.56, $SD$ = 2.87, $\alpha$ = 0.79].

We assessed women's role overload using the revised 6-item Reilly Role Overload Scale [65]. Examples of items are, "I need more hours in the day to do all the things that are expected of me," and "There are times when I cannot meet everyone's expectations." Items were rated on a seven-point Likert scale [1 = *never* to 7 = *always*], and the average of the items was computed [$M$ = 4.39, $SD$ = 0.99, $\alpha$ = .80].

In a final step for operationalizing exposure to stressors, we computed a multiple-indicator index score [54]. Indicators first were standardized, and then summed and standardized again, to yield a composite z-score [*range* = -2.41 to 3.88; modest positive skew] with higher scores representing greater exposure to multiple chronic stressors. Indicators included the stressful life events, single parenthood, Hollingshead SES [reversed], lack of social support, CBCL externalizing problems, ECBI intensity scale, the child misbehavior scale, parenting daily hassles, household chaos, and role overload [57].

**Sleep.**  To measure sleep, mothers wore a watch-like actigraph on their non-dominant wrist. The actigraph, the MicroMini Motionlogger from Ambulatory Monitoring, Inc. [AMI; Ardsley, NY], recorded minute-by-minute patterns of motor activity. Actigraph data were scored with the Motionlogger Analysis Software Package Action W-2 software [version 2.6.92] from Ambulatory Monitoring, Inc. The Cole-Kripke algorithm, which has been validated for adults and shown to provide reliable estimates of sleep indexes when averaged over seven nights, was used to reduce the motion data into meaningful sleep variables [66, 67]. Mothers also completed a daily sleep diary to record bedtime, night wakings, and rise times. Minutes asleep while in bed were based on the bedtime reported in the daily diary and actigraphically-determined sleep end [i.e., time awake]. Variables concerning activity during the sleep period and awakenings after sleep onset were based on motion recordings by the actigraph using a moderate sensitivity threshold [68]. Night waking was scored when the activity count was above threshold for at least five minutes.

Of the total sample, 91.7% of the mothers [$N = 221$] had at least five nights of usable actigraph data, which is in accordance with guidelines as the number of nights needed to yield reliable estimates [67; 69]. Mothers provided on average, 6.72 days of data [$SD = 1.41$ days]. 7.1% of mothers [$N = 17$] provided less than 5 nights of data and 1.2% of mothers [$N = 3$] did not provide any actigraphy data.

Based on a principal components analysis [PCA] with oblique rotation, four overarching sleep components were identified based on the actigraphic and diary variables aggregated across the week of data collection. Four composite variables were created based on the highest loading variables for each component- 1] sleep duration, 2] sleep variability, 3] sleep activity, and 4] sleep timing [57; 70]. The composite representing sleep duration is composed of the mean of z-scored [standardized] actigraph variables including average time the mother spent in bed each night, the time the mother spends in bed after sleep onset, and the time the mother spent asleep each night [$M = -.05$, $SD = .86$]. The composite representing sleep variability is composed of the mean of standardized actigraph variables including the night-to- night standard deviations of: time of sleep onset, duration of time spent in bed at night, time spent asleep at night not including night wakings, and time at which the mid-point of sleep occurred [$M = .06$, $SD = .85$]. The composite representing sleep activity is composed of the mean of z-scored actigraph variables including the average time awake after sleep onset, the average of the standard deviation of minute-to-minute activity level, the average number of night wakings lasting at least five minutes, the average duration of the longest wake episode after sleep onset, and the average percent of active epochs after sleep onset [$M = -.07$, $SD = .83$]. The composite representing sleep timing is composed of the mean of z-scored actigraph variables including the average time of midsleep, average time of sleep onset, and the average bedtime reported on the sleep diary [$M = .01$, $SD = .96$]. These four sleep components explained 82% of the variance in 17 actigraphy variables and were used in all subsequent analyses. The four sleep composites demonstrated strong internal consistency [average $\alpha = .92$, ranging from .89 to .93 across composites].

**Cognitive performance.**  We measured cognitive performance using an executive function [EF] task battery and a short IQ test. We measured EF with four tasks that measured aspects of attention/set shifting, inhibitory control, and working memory [71]. Mothers completed three of the four standard EF tasks on a desktop or laptop computer: Tower of Hanoi [72], Wisconsin Card Sort [73], and Stroop Color-Word [74]. They also completed a backward digit span task, while face to face with a research assistant who recorded their responses on a score sheet. Tower of Hanoi involved moving three disks of different sizes to a target peg keeping the original order, using two rules: only one disk could be moved each turn, and larger disks could not be placed on smaller disks. Time to completion was used as the score for the

task, $M$ = 33.17 secs, $SD$ = 16.89 secs. For the Wisconsin Card Sort task, mothers were presented with four stimulus cards with different colors, quantities, and shapes and were asked to match a stack of cards to the original stimulus cards according to a matching rule [i.e., either by color, quantity, or shape]. The matching rule changed without warning, and the participant had to infer the new rule based on feedback from the computer regarding correct and incorrect responses. We used total number of correct trials [reflected, log transformed, and again reflected to transform to reduce skew], $M$ = 51.90 trials, $SD$ = 7.32 trials. For the Stroop task, mothers selected among keyboard keys representing various colors. The task involved four blocks of 20 trials each: for block 1, mothers were asked to select the color key corresponding to the name of the color written in black ink; for block 2, they were asked to select the color key corresponding to the ink color of the matching color word [i.e., congruent condition]; for block 3, they were asked to select the color key corresponding to the ink color of a nonmatching color word [i.e., incongruent condition]; and finally, for block 4, they were asked to select the color key based on the ink color of a matching or nonmatching color word [i.e., mixed congruent and incongruent condition]. We used the percent of correct responses during the mixed congruent and incongruent trial block [the fourth block], $M$ = 96.52%, $SD$ = 10.21%. For backward digit span, an experimenter read a random series of single-digit numbers [0–9] and the participant attempted to reproduce the sequence in reverse. Each participant was given two chances to correctly reproduce the sequence, and the task ended when the participant got two consecutive sequences wrong. We used the highest sequence length correctly completed, $M$ = 5.86 digits, $SD$ = 1.36 digits. These tasks and scores do not have widely established norms, but the distributions were similar to other recent studies of adult women in community samples [e.g., 30, 75].

Time to completion from the Tower of Hanoi was reverse-scored so that higher scores corresponded with better EF task performance. A principal components analysis with a forced single component solution was executed [40% explained variance, loadings from .49 to .72]. The four standardized indicators were averaged and standardized again to yield a widely and normally distributed EF z-score. If mothers had scores on two of the four EF tasks, their data was used in the EF composite. Eight participants were excluded from analysis because they had only one task score.

The Shipley IQ test [76] also was administered. The vocabulary task [40 items] had participants read a target word and then select the most synonymous of four other words. The abstract reasoning task [20 items] involved filling a blank space in an increasingly complex sequence of numbers or letters to complete a pattern in each sequence. Each item was scored 0 = *incorrect* or 1 = *correct*. Because we were interested in age differences, we used the raw total score instead of age-normed standard scores: correct vocabulary items + [2*correct abstract reasoning items]; maximum possible score = 80; $M$ = 63.79, $SD$ = 9.63, *range* = 25 to 79 [mean t-score = 105, range of t-scores from 67–121]. The Shipley and EF composite scores were moderately correlated, $r$ = .49. We standardized the Shipley score, then averaged it with the EF z-score and standardized again to compute an overall cognitive performance z-score that was widely and normally distributed.

## Results

Descriptive statistics and correlations are shown in Tables 1 and 2. There was significant covariation among greater sleep variability, later timing of sleep, shorter sleep duration, and having more stressors. Greater sleep activity also was associated with shorter sleep duration. Cognitive performance scores were lower for women with shorter sleep duration, and more variable and active sleep. To create two groups of women based on age, we categorized women

**Table 1. Descriptive statistics.**

| | *M* | *SD* |
|---|---|---|
| Average time spent in bed [mins] | 483.98 | 53.78 |
| Average time in bed after sleep onset [mins] | 405.77 | 66.59 |
| Time spent asleep [mins] | 453.87 | 53.86 |
| Time of sleep onset [SD] | .64 | .38 |
| Time spent in bed [mins; SD] | 63.33 | 32.48 |
| Time of midsleep [SD] | .46 | .25 |
| Average time awake after sleep onset [mins] | 44.76 | 37.05 |
| Average min-to-min activity level [SD] | 36.10 | 13.42 |
| Average no. of night awakenings | 2.39 | 1.57 |
| Average duration of longest wake episode [mins] | 18.63 | 14.68 |
| Average % active epochs | 46.35 | 13.48 |
| Average time of midsleep [HH:MM[1]] | 3:12 | .55 |
| Average time of sleep onset [HH:MM[1]] | 23:55 | 1:15 |
| Changes and Adjustments Questionnaire | 2.34 | 2.27 |
| Social Support Scale | 1.00 | .63 |
| CBCL-externalizing subscale | 11.86 | 7.29 |
| Eyberg Child Behavior Inventory | 103.93 | 22.79 |
| Child Misbehavior Scale | 7.31 | 3.01 |
| Parenting Daily Hassles | 40.31 | 8.39 |
| Household Chaos | 3.56 | 2.87 |
| Reilly Role Overload | 4.39 | .99 |
| Tower of Hanoi [seconds] | 33.17 | 16.89 |
| Wisconsin Card Sort [no. of correct trials] | 51.90 | 7.32 |
| Stroop [% correct] | 96.52 | 10.21 |
| Backward DigitSpan [longest sequence correct] | 5.86 | 1.36 |

*Note*. All sleep variables refer to nighttime sleep. To account for the discontinuous nature of time in bed that occurs prior to and after midnight, a value of 24 was added to all times occurring after midnight. For illustration, assume a person went to bed at the following times on a 24-hour clock for 7 nights: 22, 23, 22, 00, 02, 21, 22 hours. Without accounting for discontinuity of time at midnight, the average bedtime would incorrectly be 16 hours. Modifying the times to: 22, 23, 22, 24, 26, 21, 22 hours, results in an average bedtime of 22:52 hours. Therefore, time variables that occurred after midnight were adjusted before computing weekly means and standard deviations. All variables, except where indicated, were averaged over seven nights. *SD* = standard deviation of scores on that variable across participants; (*SD*) = standard deviation of that variable across seven nights.

with z-scores on age less than '0' as "younger" women [*N* = 121, *M* = 28.86 years, *SD* = 2.93 years, range was 21–32 years] and those with z-scores on age greater than '0' as "older" women [*N* = 120, *M* = 36.67, *SD* = 3.66, range was 33–50]. Parent age was not significantly associated with any of the sleep composites except sleep timing—older mothers tended to have earlier sleep timing, i.e., go to bed earlier. Parent age was also not correlated with scores on the cumulative stress index, however on the separate stress measures, older mothers reported more lack of social support, higher role overload, higher child externalizing problems, and higher SES. Parent scores on cognitive performance were not correlated with the stress index; however, lack of social support and lower SES were associated with lower cognitive performance scores. Also, mothers who provided less than 5 nights of data had significantly lower scores on cognitive performance compared to mothers who provided more than 5 nights of data, *t*(222) = -3.77, *p* < .001.

**Table 2. Correlations.**

|  | 1. | 2. | 3. | 4. | 5. | 6. | 7. |
|---|---|---|---|---|---|---|---|
| 1. Age | 1 | | | | | | |
| 2. Sleep Duration [z] | .03 | 1 | | | | | |
| 3. Sleep Variability [z] | -.06 | -.31** | 1 | | | | |
| 4. Sleep Activity [z] | -.06 | -.16* | .12 | 1 | | | |
| 5. Sleep Timing [z] | -.16* | -.55** | .36** | .03 | 1 | | |
| 6. Stressors [z] | -.09 | -.18** | .25** | .03 | .20** | 1 | |
| 7. Cognitive Performance [z] | .09 | .14* | -.16* | -.24** | -.05 | -.05 | 1 |

* $p < .05$,

** $p < .01$

To test the hypothesis regarding the link between sleep and cognitive ability and to explore the moderating effect of age and stress, we ran four hierarchical linear regressions separately for each of the four sleep composite scores. For each equation, we entered main effects at step 1[standardized values of all predictors]; at step 2, we entered the two-way interaction terms with stress and age [created by multiplying the standardized values of sleep, stress, and age]; and at step 3, we entered the three-way interaction term created by multiplying the standardized values of the sleep composite, stress, and age. Two of the four equations yielded significant effects: sleep activity and sleep timing.

## Sleep activity

The equation for sleep activity was significant, $F[7, 222] = 3.72$, $p < .001$, $R^2 = .11$. There was a significant main effect of sleep activity [$\beta = -.21$, $p = .002$] and a significant three-way interaction between sleep activity, stress, and age [see Table 3]. To interpret the three-way interaction, we used the Johnson-Neyman technique [77] separately for younger versus older women (median split on age). This technique allowed us to examine the regions where the slope between sleep activity and cognition as a function of stress was statistically significant. For younger women [see Fig 1a], the simple slopes did not change across levels of stress. For older women [see Fig 1b], the association between sleep activity and cognition was positive at low levels of stress. However, at average and high levels of stress, the association between sleep activity and cognition was negative and significant.

## Sleep timing

In the sleep timing equation, the three-way interaction term between sleep timing, stress, and age was significant, $\beta = 0.23$, $p = .01$ [see Table 4]. To interpret the three-way interaction, we

**Table 3. Hierarchical multiple regression equation predicting cognitive performance from sleep activity.**

|  | B | S.E. | β | T | p |
|---|---|---|---|---|---|
| Sleep Activity | -.21 | .07 | -.21 | -3.1 | .002 |
| Age | .10 | .07 | .10 | 1.48 | .141 |
| Stress | -.02 | .07 | -.02 | -.31 | .756 |
| Activity X Age | .11 | .07 | .11 | 1.56 | .119 |
| Activity X Stress | -.12 | .07 | -.11 | -1.70 | .091 |
| Stress X Age | -.002 | .06 | -.002 | -.04 | .970 |
| Activity X Stress X Age | -.16 | .06 | -.19 | -2.72 | .007 |

(A) (B)

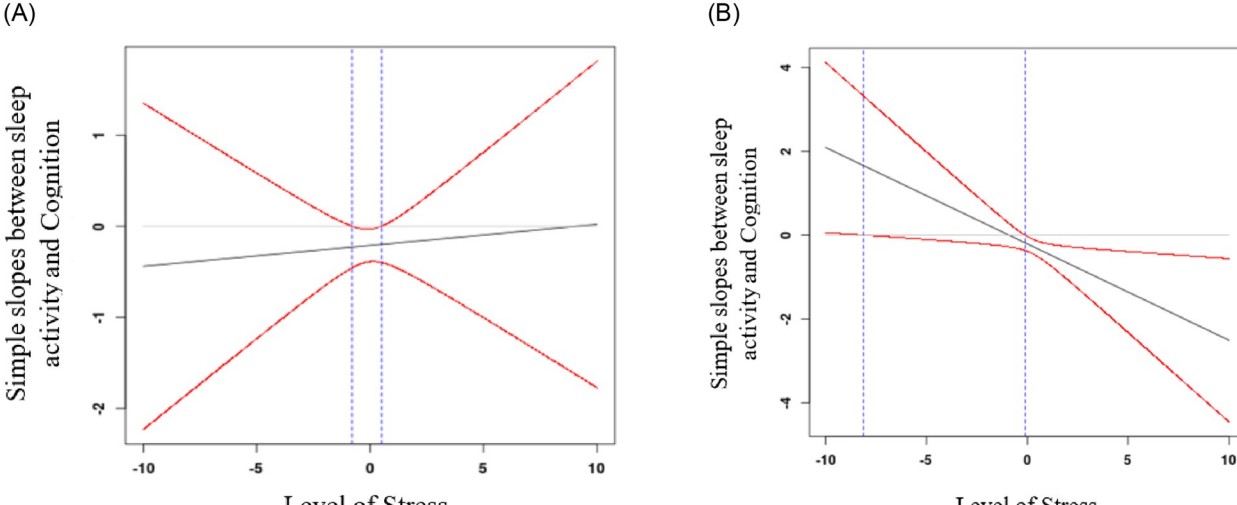

**Fig 1. a.** Johnson-Neyman regions of significance showing that for younger mothers, the effect size of the association between sleep activity and cognition do not vary much across levels of chronic stress. Red lines represent confidence intervals. Area in between the blue lines represent where the confidence interval includes zero, thus, the region where simple slopes between sleep timing and cognition are non-significant. Area outside the blue lines represent significant slopes. **b.** Johnson-Neyman regions of significance showing that for older mothers, the effect size of the association between sleep activity and cognition is significant and strongly negative for those experiencing high levels of stress. Red lines represent the confidence intervals. Area in between the blue lines represent where the confidence interval includes zero, thus, the region where simple slopes between sleep timing and cognition are non-significant. Area outside the blue lines represent significant slopes.

again used the Johnson-Neyman technique separately for younger versus older women. For younger women [see Fig 2a], at lower levels of stress, the relationship between sleep timing and cognition was positive and significant; later bedtimes were associated with better cognitive performance. For younger women with higher levels of stress, the relationship between sleep timing and cognition was negative and significant; later bedtimes were associated with poorer cognitive performance. For older women [see Fig 2b], there were no significant associations between sleep timing and cognition regardless of level of stress.

## Sleep duration and variability

For the other two sleep variables, neither contributed to a significant predictive equation: duration, $F[7, 223] = 1.11$, $p = .356$, $R^2 = .04$; variability, $F[7, 223] = 1.72$, $p = .106$, $R^2 = .05$. However, two patterns are noteworthy. First, there were significant or trending towards significant main effects of two sleep variables in the prediction of better cognitive performance: longer durations, $\beta = .13$, $p = .06$, and lower variability, $\beta = -.18$, $p = .017$. Second, in all four equations, stressor level was unrelated to cognitive performance [$\beta$s from -.02 to .007, $ps > .7$].

**Table 4. Hierarchical multiple regression equation predicting cognitive performance from sleep timing.**

|  | *B* | *S.E.* | *β* | *T* | *p* |
|---|---|---|---|---|---|
| Sleep Timing | -.06 | .07 | -.06 | -.90 | .368 |
| Age | .03 | .07 | .03 | .46 | .644 |
| Stress | -.01 | .07 | -.01 | -.06 | .949 |
| Timing X Age | .02 | .08 | .02 | .20 | .842 |
| Timing X Stress | .07 | .08 | .06 | .82 | .415 |
| Stress X Age | -.12 | .08 | -.13 | -1.59 | .114 |
| Timing X Stress X Age | .21 | .08 | .23 | 2.58 | .010 |

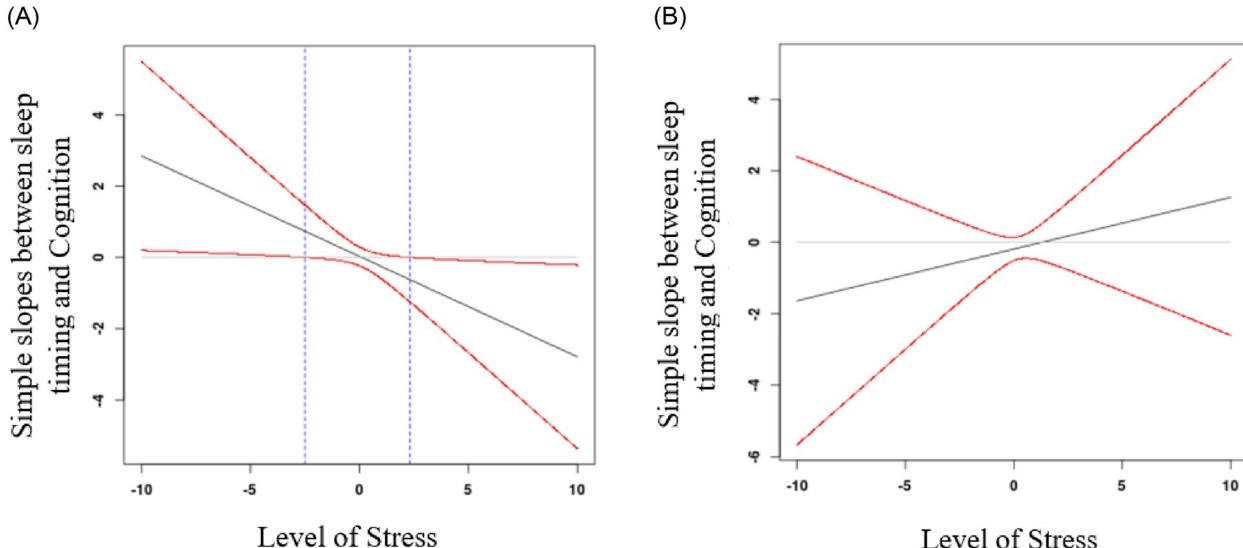

**Fig 2. a.** Johnson-Neyman regions of significance showing that for younger mothers, the effect size of the association between sleep timing and cognition is positive at low levels of stress and negative at high levels of stress. Red lines represent the confidence intervals. Area in between the blue lines represents where the confidence interval includes zero, thus, the region where simple slopes between sleep timing and cognition are non-significant. Areas outside the blue lines represent significant slopes. **b.** Johnson-Neyman regions of significance showing that for older mothers, the effect size of the association between sleep timing and cognition do not vary much across levels of chronic stress. Red lines represent the confidence intervals. Absence of blue lines indicate that none of the slopes are significant.

## Discussion

Separate literatures show that higher levels of stress and poorer sleep are each linked with diminished cognitive performance. More research is needed to examine the intersection of these two factors in explaining individual differences in cognition [8]. Mothers of toddlers are an understudied population in cognition research who experience both higher levels of stress and poorer sleep. To that end, in the current study we examined whether the link between cognitive performance and cumulative stress [due to single parenthood, child behavior problems, socioeconomic risk, low levels of social support, stressful life events, household chaos, and daily parenting hassles] in mothers of 2.5-year-olds would depend in part on various facets of maternal sleep including duration, variability, activity, and timing [hypotheses 1 and 2]. If there was an interaction effect between stress and sleep, we also planned to explore whether this moderation effect differed for younger versus older women [hypothesis 3].

Multiple regression analyses revealed that stress and the sleep facets did not independently predict variance in maternal cognitive performance [no support for hypothesis 1], suggesting that stress effects on cognition are conditioned on level of sleep activity/variability/timing/duration, i.e., the association between stress and cognition differs depending on mother's level of sleep [e.g. high versus low activity]. There was an interaction effect between stress, sleep activity and age [support for hypotheses 2 and 3]. For younger women, post-hoc probing using the Johnson-Neyman method showed that the link between stress and executive function was not moderated by sleep activity, i.e., they performed similarly well on the tasks regardless of how much sleep activity they experienced. In contrast, for older women there was a moderating effect of sleep activity [support for hypothesis 3]. Older women who had both high levels of stress and high levels of sleep activity performed substantially less well on the cognitive tasks compared to all other subgroups of women. The pattern suggested that motor activity, which we are interpreting as an index of sleep fragmentation, may be particularly likely to

impair cognitive regulation and functioning in the face of chronic stressors, especially among older women.

There was also an interaction effect between stress, sleep timing, and age [support for hypotheses 2 and 3]. For younger women, the Johnson-Neyman plots suggested that the association between sleep timing and cognition functioned differently depending on the level of stress. For young mothers with low stress, the relationship between sleep timing and cognition was positive, i.e., later bedtimes were associated with higher cognitive performance. In comparison, for younger mothers with higher levels of stress, later bedtimes were associated with poorer cognitive performance. Regarding this somewhat surprising finding for lower-stress younger women, there is one study suggesting that higher SES is associated with later bedtimes in parents [78]. The lower-stress younger mothers in our sample also were most likely to be higher SES (since SES indicators were included in the stress index), so this may have something to do with this surprising finding—although we do not have an explanation to offer at this time. In contrast, at higher levels of stress, younger mothers with a later bedtime showed poorer cognitive performance, suggesting that going to bed later in combination with living with high levels of psychosocial stress may impair cognitive performance. Among older women, there was no moderating effect of stress on the relationship between sleep timing and cognition, i.e., the mothers performed similarly on the cognitive performance tasks regardless of whether they had early or late bedtimes. This suggests that older mothers may not benefit from nor be negatively affected by going to bed at earlier or later times in terms of cognitive performance across a range of chronic stress levels.

Exposure to chronic stressors is thought to affect cognitive performance through changes in the hypothalamic-pituitary-adrenal axis [HPA] and hippocampal volume [2, 79]. High levels of HPA activity and continual dysregulation of this network of physiologic mediators lead to increased "allostatic load" on the brain, which results in impairments in learning and higher order cognitive processes. Daily stressors can also produce these transient effects by reducing the amount of attentional resources available for information processing [80]. During normative aging, these effects only become more deleterious on cognitive performance [81].

Sleep is another strong predictor of cognitive functioning. Through tissue restoration and metabolite clearance, sleep aids in consolidating memories and improving executive and attentional control [82]. Increased nighttime wakings and short sleep durations are associated with poorer executive function performance such as working memory, inhibitory control, and attentional control [83].

Mothers of toddlers are one subgroup of the population that experience both chronic high levels of stress and poorer sleep [12, 57]. Because stress has been implicated in sleep quality and quantity, with high levels of cortisol being an indicator of both higher stress and poorer sleep [84], further research is warranted to examine the interplay of these two variables in mothers and their parenting behavior. There are many sources of stress for parents of young children such as, but not limited to, lack of social support, hard-to-manage child behavior, chaotic household, low socioeconomic status, stressful life changes, and daily parenting challenges. Each of these chronic stressors may have individual effects on parenting, and it is also likely that the number of these stressors and the severity of them cumulatively predict larger individual differences in detrimental outcomes [54]. Exposure to multiple risk factors worsens outcomes more than exposure to merely one risk factor. Thus, in this study, we examined how a *cumulative* stress risk index composed of these common parenting stressors was associated with cognitive performance in a sample of mothers of 2.5-year-olds. Because sleep and age have also been found to predict cognitive performance, we explored how this association was moderated by various facets of maternal sleep and age. We tested two competing hypotheses: 1] the anticipated stress/sleep interactive effect might be diminished for older mothers because

they have acquired more strategies to cope with such demands, or 2] the hypothesized interaction effect might be strongest for older women because of the stress regulation network and cognitive system decline that occurs during typical aging.

Our findings show support for the second hypothesis. Nighttime activity, which involves night waking, was particularly detrimental to older mothers who were experiencing a high level of chronic stressors. These mothers performed much more poorly on tasks measuring cognition, suggesting that it was the cumulative effect of higher stress, poorer sleep, and older age that was predictive of variance in cognitive performance. We also found some effect of sleep timing for younger mothers who experienced low levels of stress. In this subgroup, the mothers who went to bed later performed significantly better on the cognitive performance tasks. When younger mothers experienced high levels of stress, going to bed later predicted poorer performance. This suggests that for younger mothers, the role of sleep timing may be less straightforward than it is for older women. Going to bed at later times may not be particularly detrimental for young mothers if they experience low levels of stress; indeed, for them, a later bedtime may indicate an adaptive mechanism. However, we cannot identify what explains this effect in these analyses, so results should be interpreted with caution. Neither sleep duration nor sleep variability significantly moderated the link between stress and cognition for either age group.

A few limitations of the study must be considered when interpreting the results. First, the analyses for this paper only included data from one time point, so it was not possible to examine the bidirectional nature of the association between chronic stressors and cognitive performance across time. It is possible that mothers with poorer cognitive ability may be less able to regulate daily stress and thus, have poorer sleep. Longitudinal studies on mother cognition and sleep would be required to explore the potential temporal patterns of the covariation between stress, sleep and cognition. Second, the sleep information that we analyzed did not include daytime naps, so the results reflect only nighttime sleep patterns. Third, the study relied on mother self-report on all measures of stressors, which could result in informant bias, e.g., mothers who had poorer sleep may have been more likely to rate their children more negatively on misbehavior [85]. Alternate measures of constructs such as child misbehavior and household chaos would provide additional information. Fourth, in the current sample, 90% of the mothers were White. Since there is some work suggesting that there are racial disparities in parent and child sleep [86–89], the current results may be less generalizable to non-White samples. Fifth, this was also a relatively advantaged sample with most mothers reporting low levels of stress [only 16% of the mothers were single parents and 74% of mothers had college degrees]. While there was enough variation in the sample to test the hypotheses in this paper, further research is needed to examine a wider range of socioeconomic levels.

Nevertheless, our study used a socioeconomically diverse sample that was representative of the communities where the study was conducted. Also, apart from the self-report on stressors, other main constructs of the study, sleep and cognitive performance, were measured in more objective ways. Thus, despite some limitations, our study provides a valuable look into factors that explain individual differences in cognitive performance in mothers. No other study, to our knowledge, has examined the intersection of stress, sleep, and age as factors in mothers' cognitive performance. Mothers could be especially affected by stress, sleep, and aging, when rearing young children. We were not only able to replicate previous research that shows that exposure to chronic stress is linked with both cognitive performance and sleep, but that the link between stress and cognition is moderated by nighttime waking, sleep timing, and age. Our results and conclusions emphasize the importance of examining a variety of contextual influences that may affect cognitive performance in adults.

## Supporting information

**S1 Dataset.**
(SAV)

## Author Contributions

**Conceptualization:** Kirby Deater-Deckard.

**Data curation:** Maureen E. McQuillan, Angela D. Staples.

**Formal analysis:** Mamatha Chary.

**Funding acquisition:** Kirby Deater-Deckard, John E. Bates.

**Investigation:** Kirby Deater-Deckard, John E. Bates.

**Methodology:** Kirby Deater-Deckard, Angela D. Staples, John E. Bates.

**Project administration:** Mamatha Chary.

**Supervision:** Kirby Deater-Deckard.

**Writing – original draft:** Mamatha Chary.

**Writing – review & editing:** Kirby Deater-Deckard, Maureen E. McQuillan, Angela D. Staples, John E. Bates.

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
