## [Decision Letter · Decision Letter 0]

10 Apr 2020

PONE-D-20-03649

Mother's Sleep Deficits and Cognitive Performance: Moderation by Stress and Age

PLOS ONE

Dear Ms. Chary,

Thank you for submitting your manuscript to PLOS ONE. After careful consideration, we feel that it has merit but does not fully meet PLOS ONE’s publication criteria as it currently stands. Therefore, we invite you to submit a revised version of the manuscript that addresses the points raised during the review process.

I hope that all of the authors are well in these challenging times.

I was able to secure one review of your manuscript. I thank that reviewer for their constructive feedback on the work. As you will see, there is interest in the work. However, there are some important details that need to be clarified. These include, but are not limited to, the coding of actigraphy, data reduction for the actigraphy, and construction of the index of stress. I also think that the reliance on dichotomizing variables for visualizing interactions does a disservice to understanding the interaction effects. Re-examining the post-hoc tests for the interactions by treating the variables as continuous would be more powerful. Description of simple slopes and/or regions of significance would provide statistically rigorous conclusions.

We would appreciate receiving your revised manuscript by May 25 2020 11:59PM. To enhance the reproducibility of your results, we recommend that if applicable you deposit your laboratory protocols in protocols.io, where a protocol can be assigned its own identifier (DOI) such that it can be cited independently in the future. For instructions see: http://journals.plos.org/plosone/s/submission-guidelines#loc-laboratory-protocols

We look forward to receiving your revised manuscript.

Kind regards,

Thomas M. Olino

Academic Editor

PLOS ONE

Journal Requirements:

2. Please change "Caucasian” to “White” or “of [Western] European descent” (as appropriate).

"The study has been funded by Grants MH099437 from the National Institute of Mental Health and HD073202 from the Eunice Kennedy Shriver National Institute of Child Health and Human Development."

"NO The funders had no role in study design, data collection and analysis, decision to publish, or preparation of the manuscript."

4. Please include your tables as part of your main manuscript and remove the individual files. Please note that supplementary tables (should remain/ be uploaded) as separate "supporting information" files.

Additional Editor Comments (if provided):

Reviewers' comments:

Reviewer's Responses to Questions

**Comments to the Author**

1. Is the manuscript technically sound, and do the data support the conclusions?

Reviewer #1: Partly

2. Has the statistical analysis been performed appropriately and rigorously? 

Reviewer #1: Yes

3. Have the authors made all data underlying the findings in their manuscript fully available?

Reviewer #1: Yes

4. Is the manuscript presented in an intelligible fashion and written in standard English?

Reviewer #1: Yes

5. Review Comments to the Author

Reviewer #1: This manuscript examines the cross-sectional relationships between domains of sleep, chronic stress, and cognitive performance among 227 mothers of 2.5 year-old toddlers. Strengths include the combination of actigraphy and self-report measures to assess sleep, a comprehensive index of stress, behavioral measures of executive functioning, and a heterogeneous sample of mothers with similar-age toddlers. However, there are several concerns regarding the clarity and rationale of study hypotheses, analytic approach, and discussion of study results that temper enthusiasm.

1. The hypotheses are centered on the possibility that sleep and stress may additively supplement or multiplicatively moderate one another. However, it is not clear how the authors tested their first hypothesis of the “additive effects” of stress and sleep. Are the authors referring to the main effects included in Step 1? This approach would test the effects of each predictor (when covarying for the other predictor), whereas adding standardized values of the predictors may better reflect addictive effects if this is the primary hypotheses.

2. Given the cross-sectional study design, it is impossible to know the directionality of the proposed associations and whether cognitive functioning is a predictor of poor sleep, particularly among mothers with more stress. For instance, mothers with poorer cognitive functioning may have more difficulty regulating stress in their lives (particularly when they experience heightened levels), which may contribute to poor sleep. This possibility warrants further discussion.

3. The rationale for age as the moderator of interest related to stress and sleep in predicting cognitive functioning is unclear. As the authors discuss, the ages of mothers with similarly aged toddlers are both (theoretically) linked to cognitive performance and several of the stress indicators (e.g., education, SES). Particularly since the authors do not have clear hypotheses for age, what is the practical or theoretical significance for examining age in these associations? It would be helpful to provide more information highlighting the importance of examining age as a moderator.

4. Similarly, it is surprising that certain factors were not considered or covaried in the present study, such as number of other children in the household and occupation status. This seems particularly likely to contribute to additional parental stressors, difficulty managing role overload, SES, and/or poorer sleep.

5. There is little explanation for why the specific sleep domains of interest were selected and how they may be similarly/differently associated with cognitive performance (particularly in interaction with stress/age). Some of this is discussed later in the manuscript, and would be more helpful in the introduction.

6. Although the cumulative risk index was used in the authors’ prior study, there are several indicators that are more strongly related (e.g., SES, household status, role overload, etc) and on varying time scales (e.g., daily hassles and SES). How is the CRI calculated and has it been validated as a measure of chronic stress with these indices? For instance, it is also surprising that the CBCL externalizing symptoms scale is included.

7. The use of PCA for actigraphic sleep is an interesting approach, and utilizes more of the actigraphic data. However, it is unclear how these latent domains are similar to the widely-used and validated sleep characteristics typically derived from actigraphy?

8. What is the rationale for dichotomizing continuous variables? For instance, age is dichotomized into two groups of under/over 32, which artificially creates two groups. Further, sleep variables are dichotomized into subgroups with low and high sleep activity/timing rather than applying a median split as used for other variables.

9. What was the process for scoring actigraphy? For instance, how did these scoring metrics compare to those reported by Patel and colleagues (2015) to enhance reproducibility of actigraphy scoring?

10. For the daily hassles measure, how was this score computed? Was the stressor weighted by intensity or were these summed for a total of intensity and frequency?

11. How do the norms of cognitive performance tests in the current study compare to those in the general population? Were the cognitive tasks normed for age and gender?

12. How did mothers who were included in the current study compare to those eliminated with missing data? It seems that there are 314 in the prior published study with same sample.

6. PLOS authors have the option to publish the peer review history of their article (what does this mean?). If published, this will include your full peer review and any attached files.

Reviewer #1: No

---

## [Author Response · Author response to Decision Letter 0]

20 May 2020

Comments from Editor:

**We have made the necessary changes to section headings, tables, figures, and references.**

2. Please change "Caucasian” to “White” or “of [Western] European descent” (as appropriate).

**We have made this change. The text now reads “Ninety percent of mothers were White, 4% were Hispanic, 3% were African American, 1% were Asian American, and 2% identified as mixed race, American Indian, or other.” See page 9, line 178.

“Also, 90% of the mothers were White, rendering our results less generalizable to non-White samples” See page 24, lines 521-522.**

"The study has been funded by Grants MH099437 from the National Institute of Mental Health and HD073202 from the Eunice Kennedy Shriver National Institute of Child Health and Human Development."

"NO The funders had no role in study design, data collection and analysis, decision to publish, or preparation of the manuscript."

**We have removed the funding-related text from the Acknowledgements section of the manuscript. Instead, we have added this information to the Funding Statement section of the online submission form.**

4. Please include your tables as part of your main manuscript and remove the individual files. Please note that supplementary tables (should remain/ be uploaded) as separate "supporting information" files.

**We have included tables as part of the main manuscript. See pages 17-20.**

Comments from Reviewer:

1. The hypotheses are centered on the possibility that sleep and stress may additively supplement or multiplicatively moderate one another. However, it is not clear how the authors tested their first hypothesis of the “additive effects” of stress and sleep. Are the authors referring to the main effects included in Step 1? This approach would test the effects of each predictor (when covarying for the other predictor), whereas adding standardized values of the predictors may better reflect addictive effects if this is the primary hypotheses.

**We were referring to the main effects in Step 1. To clarify, we have removed the word “additive” from the manuscript (in multiple locations), including the hypothesis 1 wording. By “additive” we meant whether sleep and stress provided “independent” prediction of multiple predictors; therefore we have replaced “additive” with “independent” throughout the manuscript (to distinguish it from a non-independent “interactive” or “moderation” effect). We also clarified (p. 18, lines 391-394) that we included the products of standardized predictors, to test interaction effects. 

2. With the cross-sectional study design, it is impossible to know the directionality of the proposed associations and whether cognitive functioning is a predictor of poor sleep, particularly among mothers with more stress. For instance, mothers with poorer cognitive functioning may have more difficulty regulating stress in their lives (particularly when they experience heightened levels), which may contribute to poor sleep. This possibility warrants further discussion.

**We agree that the cross-sectional design does not allow us to test this hypothesis. We have added the limited scope of analyses with cross-sectional data as a limitation in the discussion (p. 23, lines 511-514), and point out that a different temporal pattern like that suggested by the reviewer, could be operating.**

3. The rationale for age as the moderator of interest related to stress and sleep in predicting cognitive functioning is unclear. As the authors discuss, the ages of mothers with similarly aged toddlers are both (theoretically) linked to cognitive performance and several of the stress indicators (e.g., education, SES). Particularly since the authors do not have clear hypotheses for age, what is the practical or theoretical significance for examining age in these associations? It would be helpful to provide more information highlighting the importance of examining age as a moderator.

**Adding age as a moderator allowed us to examine whether the link between sleep, stress and cognitive performance would operate differently for older vs. younger mothers. Since there is literature showing that age is associated with sleep differences (e.g., declines in slow wave sleep and total duration of sleep; Hall et al., 2015), and because age is associated with differences in parental stress (younger mothers often have less education and financial security; older mothers report less conflict with children and report feeling more competent when dealing with childrearing stress; Barnes 2006; Barnes, Gardiner, Sutcliffe & Melhuis, 2013), adding age as a moderator would allow us to see if the relationship between sleep and stress on cognitive performance was dependent on mother age. We have added this information in the introduction section. See pages 6-7, lines 130-134.**

4. Similarly, it is surprising that certain factors were not considered or covaried in the present study, such as number of other children in the household and occupation status. This seems particularly likely to contribute to additional parental stressors, difficulty managing role overload, SES, and/or poorer sleep.

**In the McQuillan et al 2019 paper we cite, the number of children as well as number of hours worked outside the home were included as covariates in the relationship between stress and parenting. It was found that mothers’ sleep was associated with stress, even with these covariates accounted for, so we did not include it in the stress composite in the current paper. As for occupation status, occupational prestige is included in the Hollingshead SES score that we used for the study. **

5. There is little explanation for why the specific sleep domains of interest were selected and how they may be similarly/differently associated with cognitive performance (particularly in interaction with stress/age). Some of this is discussed later in the manuscript, and would be more helpful in the introduction.

**Most research on sleep effects on cognitive performance have looked at sleep deprivation, i.e., short durations of sleep. There is research showing that aging is related to decreases in total duration of sleep and more nighttime activity during sleep. There is also work showing that daily stress is related to more sleep variability and shorter durations of sleep. We have added these studies to the introduction to rationalize the use of our sleep components. See pages 4-5, lines 84-98). 

6. Although the cumulative risk index was used in the authors’ prior study, there are several indicators that are more strongly related (e.g., SES, household status, role overload, etc) and on varying time scales (e.g., daily hassles and SES). How is the CRI calculated and has it been validated as a measure of chronic stress with these indices? For instance, it is also surprising that the CBCL externalizing symptoms scale is included.

 **We chose the stressors we used for the CRI because they were comparable to the three domains of the widely-used Parenting Stress Index (Abidin, 1990): feelings of distress, difficult child behavior, and dysfunctional parent-child interactions. Also though there may not be consensus about which stress variables should be included for a cumulative measure of stress, previous research has shown that the precise combination of stressors is not as important compared to the choice to consider risk in aggregate (Sameroff et al., 1993; Evans, Whipple & Li, 2013).

The CBCL was included since there is research showing that mother perceptions of high levels of child aggression can be a significant stressor (Miragoli et al., 2018, Child Abuse & Neglect).**

7. The use of PCA for actigraphic sleep is an interesting approach, and utilizes more of the actigraphic data. However, it is unclear how these latent domains are similar to the widely-used and validated sleep characteristics typically derived from actigraphy?

** Sleep efficiency is one commonly used actigraphy variable used in sleep research. However, we chose not to use it for several reasons. We argue that our composite variables expand on the sleep efficiency variable because they incorporate more information. For example, our sleep activity composite includes the average time awake after sleep onset, the average minute-by-minute activity level, the average number of night awakenings lasting at least five minutes, the average duration of the longest wake episode after sleep onset, and the average percent of active epochs after sleep onset. Notably, in our data, our sleep activity composite is strongly correlated with the actigraphic sleep efficiency variable, r= - 0.94, such that more activity in the night reflects less efficient sleep, as expected.

The sleep timing composite similarly includes more information as it is comprised of average time of midsleep, average time of sleep onset, and average bedtime. 

Given the high correlation between the actigraphic indexes, our findings with the sleep composites can be interpreted as findings pertaining to sleep efficiency, but our usage of the composite variable also allows our findings to be interpreted relative to other variables that are commonly used in the sleep literature. Additionally, one issue with using the sleep efficiency variable alone (as pointed out by Meltzer, Montgomery-Downs, Insana, and Walsh, 2012), is that there is little to no consensus in the field about precisely how this variable should be computed. In fact, only about 65% of the studies reviewed by Meltzer and colleagues that used “sleep efficiency” reported how this variable was defined or computed. For sleep efficiency, one could use the ratio of time asleep over the sleep period from sleep onset to offset, or one could use the ratio of time asleep over the down period/period of time in bed from diary reported downtime and uptime. Without consensus in the field, we opted to analyze our data with the more generalizable, but comparable, composite indices. Our four sleep composites—sleep duration, sleep timing, sleep variability, and sleep activity—represent broad dimensions of actigraphy that are often examined in the sleep literature (Meltzer et al., 2012).

Finally, the concern we would have with using only one or two indicators for each of the major sleep dimensions is that they are not as reliable as they could be; making full use of all available data from actigraphy and diaries, addresses this measurement concern. 

Published papers that have used our sleep composites include:

1. Staples, A. D., Bates, J. E., Petersen, I. T., McQuillan, M. E., & Hoyniak, C. (2019). Measuring sleep in young children and their mothers: Identifying actigraphic sleep composites. International Journal of Behavioral Development, 43(3), 278-285.

2. Hoyniak, C. P., Bates, J. E., Staples, A. D., Rudasill, K. M., Molfese, D. L., & Molfese, V. J. (2019). Child sleep and socioeconomic context in the development of cognitive abilities in early childhood. Child Development, 90(5), 1718-1737.

3. McQuillan, M. E., Bates, J. E., Staples, A. D., & Deater-Deckard, K. (2019). Maternal stress, sleep, and parenting. Journal of Family Psychology, 33(3), 349-359.

4. Chary, M., McQuillan, M. E., Bates, J. E., & Deater-Deckard, K. (2020). Maternal Executive Function and Sleep Interact in the Prediction of Negative Parenting. Behavioral Sleep Medicine, 18(2), 203-216.

5. Hoyniak, C.P., Bates, J.E., McQuillan, M.E., Staples, A.D., Petersen, I. T., Rudasill, K.M., & Molfese, V.J. (2020). Sleep across early childhood: Implications for internalizing and externalizing problems, socioemotional skills, and cognitive and academic abilities in preschool. Journal of Child Psychology and Psychiatry. doi: 10.1111/jcpp.13225

6. Hoyniak, C.P., Bates, J. E., McQuillan, M.E., Albert, L. E., Staples, A.D., Molfese, V.J., Rudasill, K.M., & Deater-Deckard, K. (under review). The family context of toddler sleep: Routines, sleep environment, and emotional security induction in the hour before bedtime. 

7. McQuillan, M. E., Bates, J. E., Staples, A. D., Hoyniak, C. P., Rudasill, K.M., & Molfese, V. J. (in preparation). Sustained attention across toddlerhood: The roles of language and sleep. 

8. Hoyniak, C.P., McQuillan, M.E., Bates, J. E., Staples, A. D., Schwichtenberg, A.J., & Honaker, S.M. (in preparation). Objective and subjective pre-sleep arousal and sleep in early childhood.**

8. What is the rationale for dichotomizing continuous variables? For instance, age is dichotomized into two groups of under/over 32, which artificially creates two groups. Further, sleep variables are dichotomized into subgroups with low and high sleep activity/timing rather than applying a median split as used for other variables.

**Median splits were used for all the variables including sleep to facilitate analysis using ANOVAs, which are simply a special case of the general linear model. We chose to use ANOVAs in order to more simply represent the more complex, higher-order interaction effects.**

9. What was the process for scoring actigraphy? For instance, how did these scoring metrics compare to those reported by Patel and colleagues (2015) to enhance reproducibility of actigraphy scoring?

**Whenever possible, mothers’ sleep diaries were used to inform and enhance data obtained from actigraphs, such as end of rest period. All mother actigraph data were scored with the adult-validated Cole-Kripke algorithm (Cole, Kripke, Gruen, Mullaney, & Gillin, 1992). Periods when the actigraph was not worn or when sleep occurred while moving, such as in a car, were excluded from sleep scoring so that sleep minutes would not be overestimated or underestimated.

Sleep was scored during the period between the diary reported bedtime (or nap time) and the first epoch for which the actigraphic activity count reached 50 and remained above that threshold until the next sleep interval. Minutes asleep while in bed were based on the bedtime reported in the diary and actigraphic measure determined end of sleep. Variables concerning activity and awakenings after sleep onset were based on motion recordings by the actigraph using the zero-crossing mode and a moderate sensitivity threshold, which is the commonly used method in sleep research (Meltzer, Montgomery-Downs, Insana, & Walsh, 2012). A night waking was scored when the activity count was above threshold (50 crossings) for at least five minutes. The actigraph variables considered in this study were consistent with those variables used in prior research with adults (Berger et al., 2008; Meltzer et al., 2012a).**

10. For the daily hassles measure, how was this score computed? Was the stressor weighted by intensity or were these summed for a total of intensity and frequency?

**The Parenting Daily Hassles Questionnaire is comprised of 20 items that yield two subscales: challenging child behavior (7 items) and parenting tasks (8 items). In this paper, we summed the mothers’ ratings of frequency of the eight parenting tasks (1 = never to 4 = constantly; range : 0- 32) as well as her ratings’ of the intensity of the hassle (1 = no hassle to 5 = big hassle; range: 0 – 40) for those eight tasks. Both ratings were summed together for the final score. 

We have added this information to the manuscript (p. 11, lines 230-233).**

11. How do the norms of cognitive performance tests in the current study compare to those in the general population? Were the cognitive tasks normed for age and gender?

**For the Shipley test, raw scores were used because mother age was used as a predictor in the equation. Using the age-normed scores would remove the variance due to age in the scores, and also render interpretation of any estimated interaction with age uninterpretable. However, in our sample, for the age-normed scores, M = 105.47, SD = 9.48. This is very close to the population average of 100, SD = 15 (Mason, C. F., & Ganzler, H. (1964). Adult norms for the Shipley Institute of Living Scale and Hooper Visual Organization Test based on age and education. Journal of Gerontology, 19(4), 419-424.)

Also, the mother EF tasks do not have published age norms (although we now include, in the Method section, a note that the distributions on the EF tasks were similar to those of other studies of adult women in community samples, see page 15, lines 322-324). Therefore, we used non-normed scores for both cognitive tasks.**

12. How did mothers who were included in the current study compare to those eliminated with missing data? It seems that there are 314 in the prior published study with same sample.

**The reason the McQuillan 2019 paper has a larger sample is because they used data from a pilot study they conducted separately in Bloomington, IN (this study was done to demonstrate feasibility before the multi-site study began). In that study, mothers did not complete all of the questionnaires used in this study nor did they complete the executive function tasks, so we did not include the pilot study mothers in our analysis.**

---

## [Decision Letter · Decision Letter 1]

29 Jun 2020

PONE-D-20-03649R1

Mother's Sleep Deficits and Cognitive Performance: Moderation by Stress and Age

PLOS ONE

Dear Dr. Chary,

Thank you for submitting your manuscript to PLOS ONE. After careful consideration, we feel that it has merit but does not fully meet PLOS ONE’s publication criteria as it currently stands. Therefore, we invite you to submit a revised version of the manuscript that addresses the points raised during the review process.

The reviewer who previously reviewed your work read your work again. I thank them for their attention. I also reviewed your responses and new submission. Overall, I see the work stronger I thank them for their attention. I also reviewed your responses and new submission. Overall, I see the work stronger and clearer. The reviewer highlighted several minor areas of attention that appear easily addressable. The reviewer identified an additional sensitivity test that may be of use to further strengthen the conclusions. One point that was raised in my previous comments, as well as by the reviewer, focused on the treatment of age. You make a clear case that there are changes in cognitive function with age. However, there are no strong arguments about when those age differences emerge. Thus, the recommendation to treat age continuously is to enhance the interpretations of the results, more so than to diminish the finding. Relying on a continuous age variable and relying on the Neyman-Johnson technique to identify the age at which simple slopes are significantly different would be extraordinarily helpful for the work and make conclusions even stronger. If results are identical when relying on these alternative analytic methods, a footnote would suffice. However, if the results can speak to the age at which the post hoc relationships differ, that is very important information for the field. This is a critical issue for the manuscript.

We look forward to receiving your revised manuscript.

Kind regards,

Thomas M. Olino

Academic Editor

PLOS ONE

Reviewers' comments:

Reviewer's Responses to Questions

**Comments to the Author**

1. If the authors have adequately addressed your comments raised in a previous round of review and you feel that this manuscript is now acceptable for publication, you may indicate that here to bypass the “Comments to the Author” section, enter your conflict of interest statement in the “Confidential to Editor” section, and submit your "Accept" recommendation.

Reviewer #1: (No Response)

2. Is the manuscript technically sound, and do the data support the conclusions?

Reviewer #1: Yes

3. Has the statistical analysis been performed appropriately and rigorously? 

Reviewer #1: Yes

4. Have the authors made all data underlying the findings in their manuscript fully available?

Reviewer #1: Yes

5. Is the manuscript presented in an intelligible fashion and written in standard English?

Reviewer #1: Yes

6. Review Comments to the Author

Reviewer #1: Overall, this version of the manuscript is much improved, and I thank the authors for their thoughtful responses to the suggested revisions. I did have some comments and concerns, that if addressed, might strengthen the contribution of this manuscript.

Primary concerns:

1. Dichotomizing age in the current analyses is still a concerning approach for answering the key questions of interest, particularly given that the selected age was based on the median age and not any theoretical reasoning. For example, women who are 30 may be very different than younger mothers at 21 and more similar to mothers who are 32-33 years old. Artificially separating the sample by this age does not provide particularly helpful information towards our understanding of how age impacts mothers’ stress, sleep, and cognitive performance. I strongly advise you to reconsider this approach and its implications.

2. It is surprising that the authors did not control for whether mothers had multiple siblings or include this in the stress variable. It seems likely to be a key factor that could influence parenting stress, role overload, and many of the other aspects of stress.

More minor concerns:

1. Please acknowledge limitations of the methods and measurements of stress, and consider how this might influence results in the discussion.

2. Thank you for clarifying the differences between women in this study and the prior published work. Can you provide information about sample differences between women who had complete data for actigraphy and cognitive performance and those who were excluded from analysis?

3. If the authors are unaware of the descent of participants, the following changes are encouraged: Hispanic/Latinx, African American/Black, 1% were Asian American, and 2% identified as mixed race, American Indian, or other.

4. Throughout the manuscript, please use terms of comparison (e.g., poorer vs poor; higher vs. high, etc), since these comparisons made within the sample and not absolute.

5. In the discussion, please comment on generalizability of the sample given that the sample includes an overwhelming percentage of White mothers compared to other demographic groups.

6. Please consider using the phrase “parenting daily hassles” rather than daily hassles for accuracy.

7. Thank you for providing these references for the use of PCA for the actigraphy data. It may be helpful for readers to also see some of these references; please include key references in the current study.

8. Please remove IQ from the discussion, since this is more generally referred to as EF throughout.

7. PLOS authors have the option to publish the peer review history of their article (what does this mean?). If published, this will include your full peer review and any attached files.

Reviewer #1: No

---

## [Author Response · Author response to Decision Letter 1]

18 Aug 2020

Editor Comments:

One point that was raised in my previous comments, as well as by the reviewer, focused on the treatment of age. You make a clear case that there are changes in cognitive function with age. However, there are no strong arguments about when those age differences emerge. Thus, the recommendation to treat age continuously is to enhance the interpretations of the results, more so than to diminish the finding. Relying on a continuous age variable and relying on the Neyman-Johnson technique to identify the age at which simple slopes are significantly different would be extraordinarily helpful for the work and make conclusions even stronger.

**While age is a continuous variable in our analyses, in this analysis, we were working within the framework of a three-way interaction. Thus, we had to conduct a median split on one of the two moderators (stress and age) in order to interpret the interaction effect between the other three variables. Since parent age was the most distal variable to our analyses, we conducted a median split on age to create two groups of women. 

We are not aware of how to run a Neyman-Johnson analysis with a three-way interaction. However, we did use the suggested technique to identify regions of significance in our two-way interactions. We conducted the Neyman-Johnson analysis for two groups- younger mothers and older mothers. The results were very similar to our ANOVA results- the relationship between poor sleep and cognition was negative and significant only for older mothers experiencing high levels of stress. We have included these figures showing the regions of significance for both groups as supplementary figures. We have referenced them in text, see page 19, lines 407- 411.

“Post-hoc analyses showed that the significant difference was for those older mothers who were high on both sleep activity and stressors (see Fig 1). As a check of our results, we also used the Neyman-Johnson method to identify the regions of significance (see Fig 1b and Fig 1c). Results were very similar with both techniques- older mothers had significantly lower performance scores than the other three sub-groups, whose average performance scores were very similar.” **

Reviewer 1 Comments: 

Primary concerns:

1. Dichotomizing age in the current analyses is still a concerning approach for answering the key questions of interest, particularly given that the selected age was based on the median age and not any theoretical reasoning. For example, women who are 30 may be very different than younger mothers at 21 and more similar to mothers who are 32-33 years old. Artificially separating the sample by this age does not provide particularly helpful information towards our understanding of how age impacts mothers’ stress, sleep, and cognitive performance. I strongly advise you to reconsider this approach and its implications.

**See response to editor.**

2. It is surprising that the authors did not control for whether mothers had multiple siblings or include this in the stress variable. It seems likely to be a key factor that could influence parenting stress, role overload, and many of the other aspects of stress.

** In the McQuillan et al., 2019 paper we cite, the number of children

was included as a covariate in the relationship between stress and parenting. It

was found that mothers’ sleep was associated with stress, even with this covariate

accounted for, and sleep was associated with parenting, even with this covariate

and stress controlled.**

More minor concerns:

1. Please acknowledge limitations of the methods and measurements of stress, and consider how this might influence results in the discussion.

** We do mention this as a limitation in the discussion section of the current manuscript. On page 24, lines 519-521, the text reads : “Third, the study relied on mother self-report on all measures of stressors, which could result in informant bias, e.g., mothers who had poorer sleep may have been more likely to rate their children more negatively on misbehavior (Bernstein, Laurent, Measelle, Haily & Ablow, 2012). We have added a study that shows mothers with higher rates of fatigue perceive their children’s behavior more negatively, Bernstein, Laurent, Measelle, Hailey & Ablow, 2012.” **

2. Thank you for clarifying the differences between women in this study and the prior published work. Can you provide information about sample differences between women who had complete data for actigraphy and cognitive performance and those who were excluded from analysis?

**We have added this information, namely, that mothers who provided less than 5 nights of actigraphy data and were excluded from analysis (about 9% of the sample), had lower scores on cognitive performance compared to mothers who provided 5 or more days of data. See page 16, lines 359- 361. The text now reads: “Mothers who provided less than 5 nights of data had significantly lower scores on cognitive performance compared to mothers who provided more than 5 nights of data, t(222) = -3.77, p < .001.”**

3. If the authors are unaware of the descent of participants, the following changes are encouraged: Hispanic/Latinx, African American/Black, 1% were Asian American, and 2% identified as mixed race, American Indian, or other.

**We have made this change. See page 9, lines 180-182. The text now reads: “Ninety percent of mothers were White, 4% were Hispanic/Latinx, 3% were African American/Black, 1% were Asian American, and 2% identified as mixed race, American Indian, or other.”**

4. Throughout the manuscript, please use terms of comparison (e.g., poorer vs poor; higher vs. high, etc), since these comparisons made within the sample and not absolute.

**We have made this change in wording throughout the manuscript.**

5. In the discussion, please comment on generalizability of the sample given that the sample includes an overwhelming percentage of White mothers compared to other demographic groups.

** We have mentioned this limitation. See page 24, lines 523-527. The text now reads: “Fourth, in the current sample, 90% of the mothers were White. Since there is some work suggesting that there racial disparities in parent and child sleep [Buckhalt, El-Sheik & Keller, 2007; Gellis, 2011; Patrick, Millet & Mindell, 2016; Mindell, Sadeh, Weigand, How & Goh, 2010], the current results may be less generalizable to non-White samples.”

6. Please consider using the phrase “parenting daily hassles” rather than daily hassles for accuracy.

**We have made this change throughout the manuscript. See page 4, line 71; page 11, line 230; and page 12, line 251.**

7. Thank you for providing these references for the use of PCA for the actigraphy data. It may be helpful for readers to also see some of these references; please include key references in the current study.

 **We have included the following references in text and in references**

1) McQuillan ME, Bates JE, Staples AD, Deater-Deckard K. Maternal stress, sleep, and parenting. J Fam Psych. 2019 Feb;33(3):349-59.

2) Staples AD, Bates JE, Petersen IT, McQuillan ME, Hoyniak C. Measuring sleep in young children and their mothers: Identifying actigraphic sleep composites. Int J Behav Dev. 2019 May;43(3):278-85.

8. Please remove IQ from the discussion, since this is more generally referred to as EF throughout.

**We have deleted IQ. See page 23, line 503. The text now reads: “These mothers performed three-quarters of a standard deviation below other groups on tasks measuring cognition.”**

---

## [Editor Report · Decision Letter 2]

25 Aug 2020

PONE-D-20-03649R2

Mother's Sleep Deficits and Cognitive Performance: Moderation by Stress and Age

PLOS ONE

Dear Dr. Chary,

Thank you for submitting your manuscript to PLOS ONE. After careful consideration, we feel that it has merit but does not fully meet PLOS ONE’s publication criteria as it currently stands. Therefore, we invite you to submit a revised version of the manuscript that addresses the points raised during the review process.

Thank you for your responses. I appreciate the challenges of interpreting three-way interactions when those interactions are based on continuous variables. However, methods are present to visualize and interpret the data without truncating much information. For example, http://www.jeremydawson.co.uk/slopes.htm, https://tomhouslay.com/2014/03/21/understanding-3-way-interactions-between-continuous-variables/, or (perhaps most useful) https://cran.r-project.org/web/packages/interactions/vignettes/interactions.html. Frankly, I do not think that the implementation matters; however, to strengthen the conclusions that you are presenting, relying on the data, rather than dichotomizing the continuous predictors.

At a minimum, providing support for the three-way interaction using the continuous variables is necessary. That could provide enough support for the approach taken to visualize the results.

The supplementary file with the full dataset is very helpful. However, some identification of the continuous and dichotomous variables used in the main analyses would be critical for the spirit of data sharing and transparency.

We look forward to receiving your revised manuscript.

Kind regards,

Thomas M. Olino

Academic Editor

PLOS ONE

---

## [Author Response · Author response to Decision Letter 2]

7 Oct 2020

Editor Comments:

1) I appreciate the challenges of interpreting three-way interactions when those interactions are based on continuous variables. However, methods are present to visualize and interpret the data without truncating much information. For example, http://www.jeremydawson.co.uk/slopes.htm, https://tomhouslay.com/2014/03/21/understanding-3-way-interactions-between-continuous-variables/, or (perhaps most useful) https://cran.r-project.org/web/packages/interactions/vignettes/interactions.html. Frankly, I do not think that the implementation matters; however, to strengthen the conclusions that you are presenting, relying on the data, rather than dichotomizing the continuous predictors.

At a minimum, providing support for the three-way interaction using the continuous variables is necessary. That could provide enough support for the approach taken to visualize the results.

**We appreciate the editor’s emphasis on using continuous variables wherever possible. However, there are a few reasons why we were unable to use this method for our analyses:

a) While there are certainly templates online (such as the links provided by the editor) to visualize 3-way interactions, all of the formats dichotomize at least one of the moderators into two groups and graph out the interaction effect between the continuous independent variable and the continuous dependent variable as a function of the continuous proximal moderator. There is no way that we know of that allows us to present a graph with all three of the variables remaining continuous. As quoted on the DataScience website regarding graphing interaction effects, “Three-way interactions between continuous variables create a 4D surface between all continuous variables and the response variable. [sic] Human mind cannot grasp 4D surface so we have to rely on simplifications (similar in ways to what can be done for two-way interactions) to explore three-way interactions."

b) The other reason why we chose to not use the templates provided in this review is because in a previous review letter, the editor had suggested using the Johnson-Neyman technique to identify regions of significance in interaction effects. This technique is the most robust method to leverage the value of continuous variables. In the graphs (figures 1a, 1b, 2a, and 2b) the lines represent the effect size of the relationship between the independent and dependent variable at various levels of the moderator. This way, instead of examining just the means as one would with a categorical variable, we are taking into account how the association between the two continuous variables changes along values of the continuous moderator. 

In sum, we agree with the editor that we need to leverage the value of continuous variables. Upon further reflection, we agree that the Johnson-Neyman technique is thus far, the most accurate and rigorous way to probe the post-hoc analyses of a three-way interaction. Thus, we have chosen to re-analyze all of the interaction effects using this method as opposed to the ANOVA we had previously used. We found that the pattern of results remained the same when using the Johnson-Neyman plots for sleep activity. For sleep timing however, there was a change in which regions were significant. We find now that sleep timing interacts with stress and age on cognition only among younger women. The results are explained in detail in our results section (see page 19) and our discussion (see pages 21-22).**

2) The supplementary file with the full dataset is very helpful. However, some identification of the continuous and dichotomous variables used in the main analyses would be critical for the spirit of data sharing and transparency.

**We have attached a dataset that only includes the variables used in this paper. The column “Label” explains what each variable is.**

---

## [Editor Report · Decision Letter 3]

12 Oct 2020

Mother's Sleep Deficits and Cognitive Performance: Moderation by Stress and Age

PONE-D-20-03649R3

Dear Dr. Chary,

We’re pleased to inform you that your manuscript has been judged scientifically suitable for publication and will be formally accepted for publication once it meets all outstanding technical requirements.

Kind regards,

Thomas M. Olino

Academic Editor

PLOS ONE

Additional Editor Comments (optional):

I appreciate the use of the J-N method to clarify the interaction effects found. Thank you for your contribution!
---

## [Editor Report · Acceptance letter]

14 Oct 2020

PONE-D-20-03649R3 

Mothers’ Sleep Deficits and Cognitive Performance: Moderation by Stress and Age 

Dear Dr. Chary:

I'm pleased to inform you that your manuscript has been deemed suitable for publication in PLOS ONE. Congratulations! Your manuscript is now with our production department. 

Kind regards, 

on behalf of

Dr. Thomas M. Olino 

Academic Editor

PLOS ONE